# OpenReview forum: "CVE-Bench: A Benchmark for AI Agents’ Ability to Exploit Real-World Web Application Vulnerabilities"
_ICML.cc/2025/Conference — ICML 2025 spotlightposter_

### Official Review · Reviewer_aA1n · 2025-02-23

**Overall Recommendation:** 4

**Summary:**

This paper presents CVE-Bench, a new benchmark designed to evaluate AI agents in the cybersecurity domain, specifically focusing on real-world web vulnerabilities. It compiles 40 CVEs from the past year, covering eight attack types, to create a comprehensive assessment framework. To simulate realistic exploitation scenarios, CVE-Bench incorporates two evaluation settings - zero-day (no prior vulnerability information) and one-day (high-level vulnerability description provided). The benchmark includes an automatic evaluation framework to assess AI agents' performance in exploiting these vulnerabilities. Three cybersecurity agents—CyAgent, T-Agent, and AutoGPT—are evaluated on CVE-Bench using one LLM (OpenAI GPT-4o). The study provides both quantitative and qualitative analyses, detailing success rates, failure modes, and limitations of current AI-driven cybersecurity frameworks on this benchmark.

**Claims And Evidence:**

I found the claims to be made sound.

**Essential References Not Discussed:**

The paper does not sufficiently discuss existing cybersecurity benchmarks, despite their relevance to the study. While CyBench is mentioned, CTFs are widely explored in InterCode-CTF [Yang et al., 2023] published in NeurIPS 2023, and NYU-CTF Bench [Shao et al., 2024] published in NeurIPS 2024. Meta’s CyberSecEval 1 , 2 [Bhatt et al., 2023; 2024] and CyberSecEval 3 [Wan et al., 2024] benchmarks are also publicly available. These benchmarks assess LLMs' ability in vulnerability exploitation, detection, and security-related reasoning, making them directly relevant to the paper’s contributions.

**Experimental Designs Or Analyses:**

Overall, I found the design of the benchmark good and the experimental designs and analyses are sufficient.
I think that the following could make the paper even better:
1. Using the same environment for each agent - for example, install sqlmap in the container of CyAgent and not only for the T-Agent.
2. Evaluating additional LLMs (e.g., Claude, Gemini, open-source models like LLaMA or Mistral) would provide a broader understanding of model capabilities and generalization on this benchmark. I know that this takes time and effort so will not be judging the paper negatively for this.

**Methods And Evaluation Criteria:**

The methods and evaluation on the new benchmark seems realistic.

**Other Comments Or Suggestions:**

The top bar title needs to be changed from “Submission and Formatting Instructions for ICML 2025”.

**Other Strengths And Weaknesses:**

Strengths:
1) The paper is well-written, self-contained, and easy to follow, making it accessible even to readers without a cybersecurity background. It effectively explains the different aspects of the benchmark.
2) The benchmark is designed to simulate real-world web application vulnerability exploitation, ensuring practical relevance. The methodology used to construct the benchmark is well-founded.
3) The introduction of an automated evaluation system makes the benchmark reproducible, allowing other researchers to test their AI agents on the same framework.
Weaknesses:
1) The benchmark does not contain “test” and “development” splits, making it hard for correct evaluations of AI applications on top of it in the future.

**Questions For Authors:**

1) Do you have any quantitative results regarding the failure modes described in Section 4.3?
2) In Section 4.2 (Exploit Composition), it is stated:
“With sqlmap, T-Agent can locate the vulnerability and perform SQL injection automatically. According to the reasoning traces, Cy-Agent attempts sqlmap for most of the CVEs. Appropriate use of sqlmap can significant improve the success rate of exploit SQL injection vulnerabilities, …”.
Could you clarify the difference between T-Agent and Cy-Agent in their usage of sqlmap? Does Cy-Agent have access to sqlmap but fail to use it effectively, or is the tool configured differently across the agents?
3) In Figure 3, T-Agent performs better than AutoGPT in the zero-day scenario, while AutoGPT outperforms T-Agent in the one-day scenario. Do you have an explanation for this behavior?

**Relation To Broader Scientific Literature:**

This paper introduces a new benchmark for finding CTF exploits.

**Theoretical Claims:**

N/A

---

> ### Author Rebuttal · Authors · 2025-03-30
>
> We thank the reviewer for their insightful comments. We will incorporate the suggestions in the revision.
>
> > **E1**: Using the same environment for each agent - for example, install sqlmap in the container of CyAgent and not only for the T-Agent.
>
> We clarify that sqlmap is installed in the container of both Cy-Agent and T-Agent. We will revise the experiments to provide the same set of tools for all agents.
>
> > **E2**: Evaluating additional LLMs (e.g., Claude, Gemini, LLaMA, and Mistral) would provide a broader understanding of model capabilities and generalization on this benchmark. I know that this takes time and effort so will not be judging the paper negatively for this.
>
> We find that the open-source model, Llama 3.1, achieves only 0% success rates in the zero-day or one-day settings. We will run experiments with more LLMs in the revision, such as Claude.
>
> > **Q1**: Do you have any quantitative results regarding the failure modes described in Section 4.3?
>
> We summarized the frequency of the failure modes as follows. Agents can fail due to multiple reasons, therefore, the sum of the frequency of all failure modes can exceed 100%.
>
> |                           | zero-day |         |         | one-day  |         |         |
> |---|---|---|---|---|---|---|
> |                           | Cy-Agent | AutoGPT | T-Agent | Cy-Agent | AutoGPT | T-Agent |
> |Limited Task Understanding | 30.0%    | 15.0%   |    0%   | 20.0%    |  5.0%   |    0%   |
> |Incorrect Focus            |    0%    |    0%   | 35.0%   |    0%    |    0%   | 30.0%   |
> |Insufficient Exploration   | 67.5%    | 72.5%   | 80.0%   | 37.5%    | 45.0%   | 55.0%   |
> |Tool Misuse                | 47.5%    |  5.0%   | 17.5%   | 27.5%    | 22.5%   | 10.0%   |
> |Inadequate Reasoning       | 10.0%    |  7.5%   |  7.5%   | 40.0%    | 27.5%   | 20.0%   |
>
> As shown, all agents are bottlenecked by insufficient exploration, meaning that they failed to identify the vulnerability endpoint of applications, even when high-level vulnerability descriptions were provided. T-Agent consistently understood task targets (w/ 0% Limited Task Understanding), while it sometimes focused on websites other than the vulnerable one provided in the prompt (e.g., www.example.com). Generally, compared to the 0-day setting, agents with 1-day descriptions had a lower frequency of naive failures, including Limited Task Understanding, Incorrect Focus, and Insufficient Exploration, but failed more due to Tool Misuse and Inadequate Reasoning.
>
> > **Q2**: Could you clarify the difference between T-Agent and Cy-Agent in their usage of sqlmap? Does Cy-Agent have access to sqlmap but fail to use it effectively, or is the tool configured differently across the agents?
>
> We clarify that both T-Agent and Cy-Agent have access to sqlmap with the same configuration. However, they have different approaches to use sqlmap.
> - T-Agent uses a hierarchical structure with a team manager to determine the timing to use sqlmap and has a specialized agent dedicated specifically to SQL injection attacks.
> - Cy-Agent uses sqlmap as a general-purpose tool without a specialized framework for SQL injection.
>
> We find the hierarchical planning and task-specific agents in T-Agent enhance its ability to use tools effectively, compared to Cy-Agent.
>
> > **Q3**: In Figure 3, T-Agent performs better than AutoGPT in the 0-day scenario, while AutoGPT outperforms T-Agent in the 1-day scenario. Do you have an explanation for this behavior?
>
> With further analysis, we found that AutoGPT can uncover new vulnerabilities that were not included in the CVE description. This occurs when AutoGPT is unable to exploit the specified vulnerabilities in the one-day scenario, but the web application contains an alternative, more exploitable vulnerability.
>
> For example, in CVE-2024-36779, the one-day description targets a SQL injection vulnerability in `editCategories.php`, requiring a complex, time-based blind SQL injection. AutoGPT struggled with this but uncovered a vulnerability in index.php, which could be easily exploited by using `‘OR 1=1 –` to bypass filters and gain administrator access.
>
> By identifying easier vulnerabilities in the one-day setting, AutoGPT achieved 4.5% and 5.0% higher success rates with one or five attempts, respectively, while T-Agent didn’t find new vulnerabilities. Nonetheless, when focusing solely on the described vulnerabilities, T-Agent continues to outperform AutoGPT in the one-day scenario.
>
> We will add this explanation as a case study in the revision.
>
> > The paper does not sufficiently discuss existing cybersecurity benchmarks, despite their relevance to the study.
>
> Thank you for suggesting relevant CTP benchmarks and Meta's CyberSecEval benchmarks. We will add and discuss them in the related work section of the revision.
>
> > The top bar title needs to be changed from “Submission and Formatting Instructions for ICML 2025”.
>
> We will fix the formatting issues in the revision.

---

### Official Review · Reviewer_746a · 2025-03-13

**Overall Recommendation:** 3

**Summary:**

1. This paper proposed a new benchmark for LLM-Agent Attacking.
2. Some experiments are conducted.

**Claims And Evidence:**

**Yes**

**Essential References Not Discussed:**

The authors have discussed the essential related works in Section 2. As far as I know, there is no more necessary reference.

**Experimental Designs Or Analyses:**

**Yes**

The authors conduct experiments by using three different LLM-Agents on their new benchmark. Analyses are provided in Section 4.2 and 4.3.

**Methods And Evaluation Criteria:**

**Yes**

There is no proposed method in this paper. Only the new CVE-Bench is intorduced. The authors provide a comparision with other similar benchmarks in Table 1.

**Other Comments Or Suggestions:**

There is no more comment from the reviwer.

**Other Strengths And Weaknesses:**

**Strengths:**
1. The proposed benchmark looks useful for the LLM-Agent attacking.

**Weaknesses:**
1. The  presentation of this paper should be improved.
    * (Minor) In Figure 3, the Y-axis should display "30%" rather than simply "30" to properly indicate percentage values.
    * Figure 4 lacks clarity—while the annotation mentions eight distinct tasks, only a subset appears in the chart, creating confusion for readers.
2. Despite spanning eight pages, the paper contains considerable content that appears superficial and fails to engage the reader effectively. The overall substance feels insufficient. Additional experiments and more in-depth analysis would strengthen the work and provide greater value to the research community.

**Questions For Authors:**

There is no more question from the reviwer.

**Relation To Broader Scientific Literature:**

The key contribution of this paper is the new benchmark for cyberattact -- **CVE-Bench**.

The authors have discussed the relationship bettwen the CVE-Bench and other related works such as *Cybench* and *CVE*. They are all desgined for LLM-Attacking evaluation (cyberattack).

And the broad literatures of LLM agent for attack have been discussed in Section 3.2.

**Theoretical Claims:**

**N.A.**

It looks like no theoretical claims in this paper.

---

> ### Author Rebuttal · Authors · 2025-03-30
>
> We thank the reviewer for their insightful comments. We will incorporate the suggestions in the revision.
>
> > W1: The presentation of this paper should be improved:
>
> > (Minor) In Figure 3, the Y-axis should display "30%" rather than simply "30" to properly indicate percentage values.
>
> > Figure 4 lacks clarity—while the annotation mentions eight distinct tasks, only a subset appears in the chart, creating confusion for readers.
>
> We will revise our submission to fix these presentation issues.
> 1. We will add percentage marks (%) to all the metrics calculated as percentages.
> 2. We clarify that agents didn't conduct all types of attacks successfully. Therefore, only a subset of attacks appear in the chart. We will fix the annotation to include only the successful attacks.
>
> > W2: Despite spanning eight pages, the paper contains considerable content that appears superficial and fails to engage the reader effectively. The overall substance feels insufficient. Additional experiments and more in-depth analysis would strengthen the work and provide greater value to the research community.
>
> Thank you for examining the details of our work in the appendix. We recognize the need to enhance the presentation in the appendix.
>
> In the revision, we will clean up the appendix and incorporate more comprehensive details about CVE-bench, including:
> 1. A detailed description of our data collection process
> 2. An example of exploit reproduction
> 3. An example of target containers
> 4. Running CVE-bench with standard evaluation tools, such as Inspect-AI
> 5. A sample of agent running logs

---

### Official Review · Reviewer_sgV9 · 2025-03-19

**Overall Recommendation:** 3

**Summary:**

The authors introduce CVE-Bench, a new benchmark designed to evaluate large language model (LLM) agents’ capabilities in identifying core cybersecurity vulnerabilities. They define 8 key types of core attacks that any robust system should withstand. This benchmark significantly reduces manual effort by enabling automated flaw detection within system architectures using LLM agents. The authors evaluate three different agents on CVE-Bench, providing insightful performance analysis and highlighting critical findings. The tasks are designed to reflect real-world challenges, making the benchmark highly relevant to the community. Moreover, the authors emphasize reproducibility, ensuring that others can reliably use and extend the benchmark. The difficulty of the tasks is validated through high CVSS scores, underscoring the benchmark's rigor and importance.

**Claims And Evidence:**

The authors claim that current LLM agents are not capable of solving the benchmark and provide evidence of low performance, even in a one-day setting. While their evaluation supports this claim, I am curious about the performance of SOTA web agents. For instance, web/SWE agents like OpenHands, CodiumAI (and potentially other closed sourced) could be relevant comparisons.


Additionally, I find it difficult to believe that, even when provided with a known vulnerability in the one-day setting, the agents are entirely unable to solve the tasks. This raises the question: Is the primary issue that the agents lack full contextual information about the scenario, leading them to choose the wrong tool? If so, further analysis of this failure mode would strengthen the paper’s argument.

**Essential References Not Discussed:**

The authors should consider including AgentSecurityBench (https://arxiv.org/abs/2410.02644) in their related work. This benchmark evaluates various agents on web-related attack tasks, making it a highly relevant comparison point.
I’m particularly curious to see how the three agents discussed in this paper would perform on AgentSecurityBench's tasks — and whether stronger, more capable agents exist that could provide a more comprehensive performance comparison. Including this reference could offer valuable context and help position the paper’s benchmark more effectively within the broader literature.

**Experimental Designs Or Analyses:**

Yes, the experiments conducted on the benchmark appear methodologically sound. The authors evaluate three different agents across all 8 tasks, ensuring comprehensive coverage. They also report key insights that highlight performance differences and failure patterns, contributing to a more nuanced understanding of the agents' capabilities and limitations.

**Methods And Evaluation Criteria:**

The authors evaluate their approach using standard cybersecurity testing agents, similar to those used in Cybench. The evaluation criteria are reasonable and well-aligned with the problem — they assess whether an attack succeeds using an automated grader hosted within the same container as the web application, enabling continuous monitoring.


Additionally, the authors follow established evaluation practices by running 30 iterations for GPT-4o and reporting Success@1 and Success@5 metrics. This setup seems appropriate and fair for benchmarking agent performance in this domain.

**Other Comments Or Suggestions:**

I found this typo in ‘specify’ in the CVE-2024-32980 section on page 8. Otherwise, the paper is well structured and thorough on the experiments side.

**Other Strengths And Weaknesses:**

Strengths:
1. The experiments and insights are thorough and informative, providing a clear breakdown of where each agent fails.
2. The benchmark covers a wide span of challenging tasks, which are difficult yet reproducible, making it a valuable resource for future research.
3. The evaluation setup, including Success@1 and Success@5 metrics over 30 iterations, ensures robust performance reporting.
4. The benchmark enables continuous monitoring through an automated grader, enhancing practical usability.

Weaknesses:
1. The novelty is somewhat limited. The work appears heavily inspired by CyBench, primarily extending it with more complex tasks and tools — an incremental rather than transformative contribution.
2. The authors claim that the benchmark aims to save human effort by enabling LLM agents to discover unknown vulnerabilities. However, they do not present any concrete examples where the agents uncover vulnerabilities beyond the intended tasks. This raises doubts about the benchmark’s practical utility if agents fail to explore beyond pre-defined scenarios.
3. There’s no justification or survey supporting the specific choice of agents evaluated. Including a rationale for selecting these agents — or report with other web/SWE baselines like OpenHands/Codium or — would improve the paper’s completeness and credibility.

**Questions For Authors:**

1. In the abstract, you mention that the benchmark’s use case is to identify unpredictable threats. Did any of the agents discover flaws that were previously unpredictable or unknown? If not, does this suggest that using standard testing protocols could achieve similar results, thus questioning the utility of using LLM agents for evaluation in this context?

2. Are the agents aware that the task they are performing could be harmful? Specifically, if the task violates safety guardrails, do the agents acknowledge this by saying something like, "I am sorry"? This might explain the low success rate if the agents avoid performing risky actions.

3. Could you justify why you evaluated only these three agents on your benchmark? Are there other, perhaps stronger, agents that could provide more valuable insights into the tasks?

4. When the service is provided through APIs or libraries that lack a text-based user interface, does providing more detailed information or instructions about the tools help mitigate the agents’ misuse? What might be causing the agents to choose the wrong tool or misuse it?

**Relation To Broader Scientific Literature:**

Benchmarks for agents are especially timely and relevant, given the current surge in both research and industry interest around web-based agents. With numerous web agents emerging this year, it’s crucial to have robust evaluation frameworks to assess their capabilities. This paper contributes to that need by introducing a benchmark that covers a range of cybersecurity tasks compared to existing benchmarks, making it a valuable addition to the field.

**Theoretical Claims:**

No, there are no theoretical claims in the paper.

---

> ### Author Rebuttal · Authors · 2025-03-30
>
> We thank the reviewer for their insightful comments. We provide the following clarifications and will include them in the revision.
> >Even in the 1-day setting, agents are entirely unable to solve tasks. Analysis of failure modes is suggested.
>
> We provide a quantitative analysis of failure modes in the response to Reviewer aA1n and find that failing to identify vulnerable endpoints is the key bottleneck, even in the 1-day setting. This is partially because one-day descriptions only provide high-level descriptions of vulnerabilities, while agents need to reason about specific vulnerable endpoints. We will add quantitative analysis in a revision.
> >W1: The novelty is somewhat limited. The work appears heavily inspired by and an incremental contribution of CyBench.
>
> Although CyBench and CVE-bench both focus on cybersecurity, CVEBench focuses on _realistic_ evaluation of AI agents compared to isolated capture-the-flag (CTF) exercises.
>
> *Data Collection.* CVE-bench focuses on 40 real-world web vulnerabilities with a high impact (rated "critical" by CVSS). However, the 40 tasks in CyBench have various categories and difficulties, some of which (e.g., simple input validation) don’t reflect the critical nature of current cybersecurity challenges.
>
> *Task Formulation.* Our tasks require agents to detect vulnerabilities (0-day) and exploit them to achieve attack targets (0 & 1-day). In contrast, CyBench tasks are structured as CTF exercises, which test cybersecurity skills but don’t fully reflect real-world hacking scenarios.
>
> *Evaluation.* In CVE-bench, we evaluate whether agents can impact applications, such as data breaches or DoS. CyBench, however, evaluates flag correctness, a metric that doesn’t reflect the cybersecurity risks.
> >W2,Q1: Did agents discover flaws that were unpredictable? If not, does this suggest that using standard testing tools could achieve similar results?
>
> We find that the penetration testing tool, ZAP, identified 0 vulnerability in CVE-bench. As vulnerabilities were originally detected by human experts, CVE-bench is important to assess whether AI agents can supplement human efforts. More importantly, CVE-bench evaluates the risks of agents, providing important insights for policymakers [1, 2].
>
> In addition, we found that agents can identify new vulnerabilities distinct from those in the CVE description, showing their potential to uncover unpredictable flaws. We provide a detailed example in the response to Reviewer aA1n.
>
> We will add ZAP results and a case study of new vulnerabilities.
>
> [1] UK AISI, “AI Safety Institute approach to evaluations.” https://www.gov.uk/government/publications/ai-safety-institute-approach-to-evaluations/ai-safety-institute-approach-to-evaluations
>
> [2] US AISI, “Technical Blog: Strengthening AI Agent Hijacking Evaluations.” https://www.nist.gov/news-events/news/2025/01/technical-blog-strengthening-ai-agent-hijacking-evaluations
> >W3,Q3: Including a rationale for selecting agents — or report with web/SWE baselines like OpenHands/Codium.
>
> After further experiments, we found that OpenHands identified and/or exploited 0 vulnerabilities with 5 runs using the same LLM. This is primarily because OpenHands failed to attempt different endpoints of applications thoroughly. We will add the results in the revision. In contrast, our selected agents have various capabilities relevant to cybersecurity:
> 1. AutoGPT: Selected for its versatility and generality in handling complex tasks
> 2. Cy-Agent: Designed for cybersecurity challenges, this agent has skills and tools for cyberattacks
> 3. T-Agents: It has the SOTA ability to exploit zero-day vulnerabilities, designed with the cooperation of different cybersecurity sub-domains.
>
> >Q2: Do the agents acknowledge safety violations and deny the request?
>
> We clarify that we have carefully framed tasks in an ethical context and achieved 0 denial. Agents are instructed to act as white-hat hackers with permissions granted by application owners.
> >Q4: Do more instructions on tools help mitigate misuse? What might be causing agents to choose the wrong tool or misuse it?
>
> We used the default prompts from specific agent frameworks, with slight modifications to prevent request denials and out-of-scope behaviors. To reduce the possibility of tool misuse, we provided usage for necessary tools, such as sqlmap, in the instructions.
>
> However, given the complexity of tools—sqlmap has 200+ options—agents face challenges in selecting the optimal usage. This is compounded by the difficulty in identifying web vulnerabilities, requiring agents to explore different options. We will add details of the tools’ setup in the revision.
> >Consider including AgentSecurityBench in their related work.
>
> We will include AgentSecurityBench in the revision. AgentSecurityBench focuses on attacks on AI systems (e.g., prompt injection and memory poisoning), which is orthogonal to CVE-bench.
> >I found a typo in ‘specify’ on page 8.
>
> Thank you. We will fix typos in the revision.

---

> > ### Comment · Reviewer_sgV9 · 2025-04-02
> >
> > Thank you for addressing my concerns. I strongly encourage the authors to include ZAP results and case study on newly discovered vulnerabilities in the revised paper. Additionally, incorporating the quantitative analysis of failure modes would be valuable for the agent development community, helping improve agents for cybersecurity-related tasks.
> >
> > I am satisfied with the responses and have adjusted my score accordingly.

---

### Decision · Program_Chairs · 2025-05-01

**Decision:**

Accept (spotlight poster)

**Comment:**

This paper was well received by the reviewers. Presents a useful benchmark with some interesting findings. The reviewers point out some avenues for improved clarity in presentation - I encourage the authors to incorporate those.